# stilPCR increases the effective sequencing length of Illumina targeted next-generation sequencing

Jason D. Limberis[1]*, Roland J. Nagel[2], Soumitesh Chakravorty[2], Dena Block[2], Scott Dewell[2], Alina Nalyvayko[1], Zach Howard[3], Grant Theron[4], Rouxjeane Venter[4], John Z. Metcalfe[1]

1 Division of Pulmonary and Critical Care Medicine, Zuckerberg San Francisco General Hospital and Trauma Centre, University of California, San Francisco, San Francisco, CA, United States of America, 2 Cepheid Inc., Sunnyvale, California, United States of America, 3 Division of Experimental Medicine, University of California, San Francisco, San Francisco, CA, United States of America, 4 DSI-NRF Centre of Excellence for Biomedical Tuberculosis Research, South African Medical Research Council Centre for Tuberculosis Research, Division of Molecular Biology and Human Genetics, Faculty of Medicine and Health Sciences, Stellenbosch University, Stellenbosch, South Africa

* Jason.Limberis@ucsf.edu

**Data Availability Statement:** All files are available at www.github.com/SemiQuant/stilPCR.

**Funding:** JZM, R01AI177637, National Institute of Allergy and Infectious Diseases (NIAID), https://

## Abstract

Identifying pathogens, resistance-conferring mutations, and strain types through targeted amplicon sequencing is an important tool. However, due to the limitations of short read sequencing, many applications require the division of limited clinical samples. Here, we present stilPCR (single-tube Illumina long read PCR), which allows the generation of hemi-nested amplicons in a single tube, with Illumina indexes and adapters, effectively increasing the Illumina read length without increasing the input requirements of reagents or sample. We have successfully utilized stilPCR on clinical sputum from tuberculosis patients to detect drug resistance mutations.

## Introduction

Targeted amplicon sequencing can detect resistance-conferring mutations of important human pathogens. However, the short-read limitations of Illumina sequencing–a maximum of 600bp when 300bp pair-end sequencing is done–mean regions larger than 600bp require over-lapping amplicons demanding the splitting of a sample between two or more reactions. This is often not possible with clinical samples, which are usually limited and difficult to quantify.

Here, we outline stilPCR (single-tube Illumina long read PCR). This novel PCR technique generates sequenceable, hemi-nested amplicons with Illumina indexes and adapters (the sequencing primer binding sites) in a single tube. stilPCR thus removes the need to split samples and reduces the input amount needed. The first round of PCR uses primers containing a universal tail sequence to amplify a target up to 750 nucleotides flanking each gene of interest (**Fig 1**, step 1a). In the second round of PCR, tailed gene-specific primers target a portion of the step 1 amplicon (sharing the initial reverse primer but having a unique forward primer)

www.niaid.nih.gov/. The NIAID did not and will not have a role in study design, data collection and analysis, decision to publish, or preparation of the manuscript. Cepheid provided support in the form of salaries for authors RJN, DB, SD, and SC. All authors were involved in the study design, data collection and analysis, decision to publish, or preparation of the manuscript.

**Competing interests:** RJN, DB, SD, and SC are Cepheid employees. Other authors have no competing interests. Our affiliation with Cepheid does not alter our adherence to PLOS ONE policies on sharing data and materials.

## Step 1a: Primer limited gene specific PCR

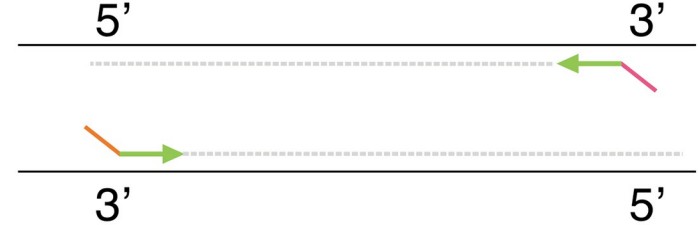

## Step 1b: Addition of indexed primers

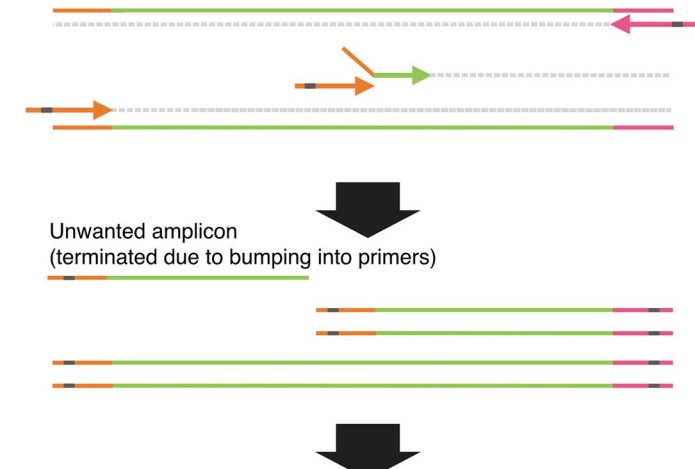

## Step 2: PCR cleanup, pooling, and sequencing

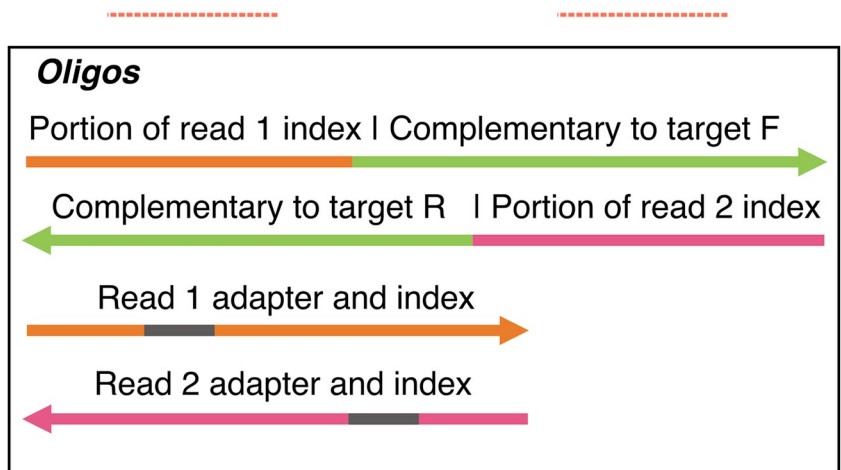

**Fig 1. Single-tube Illumina long read PCR (stilPCR) as shown for a theoretical gene.** In PCR step 1a, the full-length amplicon is generated with specific-tailed sequences at the 5'-end. After several cycles of PCR, internally binding primers are added in step 1b, resulting in the generation of several (three in this example) amplicons, all starting at the same location on the gene (shared forward primer) but ending at different locations (differing reverse primers). These amplicons are then sequenced (Step 2), allowing for coverage of the whole gene. (If pair-end sequencing is done, this

will result in relatively more coverage for the forward region of the amplicon due to its integration in all amplicons generated). F, Forward; R, Reverse.

and generate two overlapping amplicons of sequenceable length (**Fig 1**, **step 1b**) that contain the index and adapter sequence. Increasing the number of nested primers can further increase the sequenceable length. The tail sequences allow for even amplification of tailed products of different sizes. Controlling the concentrations of the gene-specific primers will allow you to produce the different amplicons at similar concentrations. The specific concentration of primer required can be determined by adding 0.5ul of 2000X SYBR green dye to the PCR reaction and monitoring the Ct values of the single plex reactions using a qPCR thermocycler. You can also control the maximum number of amplicons for Illumina cluster generation (sequencing) and ensure the maximum amount of generated data per sample by limiting our universal-tailed primers. Importantly, stilPCR-generated libraries can be pooled with any standard Illumina library, allowing for batching even with other sample types.

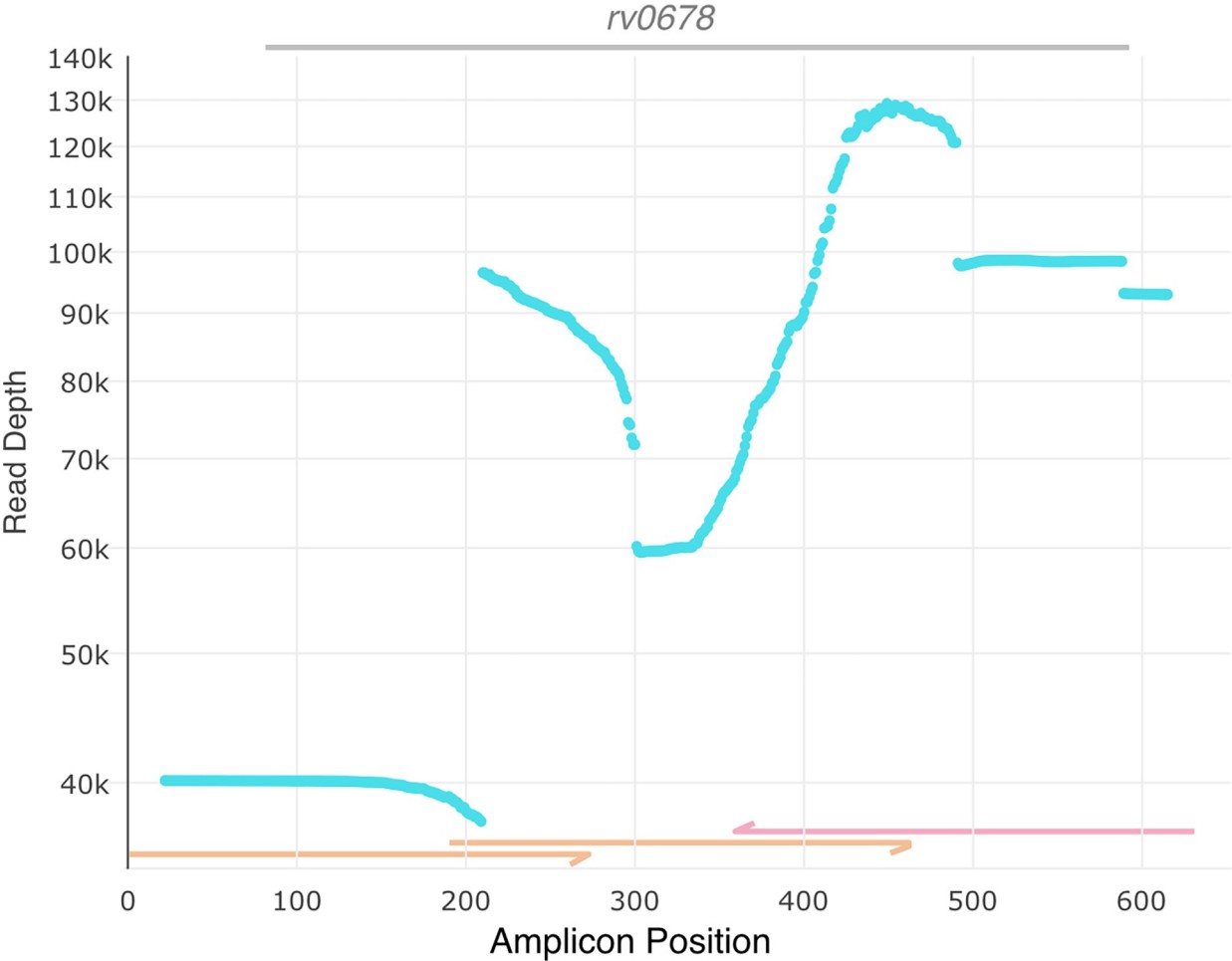

**Fig 2. Successful sequencing of *rv0678* and its promoter region using stilPCR.** The plot shows the coverage (y-axis) at each position (x-axis) along the sequence amplicons. The coding region is denoted by the grey line at the top of the figure. The positions of forward primer sequencing amplicons are shown in orange at the bottom of each plot, with reverse primer amplicons in pink.

## Materials and methods

The protocol described in this peer-reviewed article is published on protocols.io, DOI dx.doi.org/10.17504/protocols.io.bp2l6xmn5lqe/v1, and is included for printing as S1 File with this article.

## Expected results

We have successfully used stilPCR to amplify two targets, *atpE* (single primer set), *and rv0678* (stilPCR primerset [1]), of *Mycobacterium tuberculosis* in patient sputum [2]. *rv0678* is a 498bp gene and mutations along its entirety may confer *M. tuberculosis* bedaquiline resistance [3, 4]. Here, we show the expected amplicons generated for a stilPCR reaction of one gene and the successful 300bp sequencing and alignment of the stilPCR product (**Fig 2**), processed using www.github.com/SemiQuant/stilPCR. Further examples demonstrating the reproducibility of stilPCR are available in Limberis et al. [2].

## Supporting information

**S1 File. Step-by-step protocol, also available on protocols.io.**
(PDF)

## Author Contributions

**Conceptualization:** Jason D. Limberis, John Z. Metcalfe.

**Formal analysis:** Jason D. Limberis.

**Funding acquisition:** Jason D. Limberis, John Z. Metcalfe.

**Investigation:** Jason D. Limberis, Roland J. Nagel, Soumitesh Chakravorty, Dena Block, Scott Dewell, Alina Nalyvayko, Zach Howard, John Z. Metcalfe.

**Methodology:** Jason D. Limberis, Roland J. Nagel, Soumitesh Chakravorty.

**Project administration:** John Z. Metcalfe.

**Software:** Jason D. Limberis.

**Supervision:** John Z. Metcalfe.

**Visualization:** Jason D. Limberis.

**Writing – original draft:** Jason D. Limberis, Soumitesh Chakravorty, John Z. Metcalfe.

**Writing – review & editing:** Jason D. Limberis, Roland J. Nagel, Dena Block, Scott Dewell, Alina Nalyvayko, Zach Howard, Grant Theron, Rouxjeane Venter, John Z. Metcalfe.

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
