## [Decision Letter · Decision Letter 0]

27 Sep 2024

PONE-D-24-37091stilPCR increases the effective sequencing length of Illumina, targeted next-generation sequencing.PLOS ONE

Dear Dr. Limberis, 

Thank you for submitting your manuscript to PLOS ONE. After careful consideration, we feel that it has merit but does not fully meet PLOS ONE’s publication criteria as it currently stands. Therefore, we invite you to submit a revised version of the manuscript that addresses the points raised during the review process.

While the peer reviewers of your manuscript were in agreement that your described method is both novel and a worthwhile contribution to the field, they also felt that your manuscript would benefit from the clarification of certain procedural details, and the provision of details of additional validations. 

Please submit your revised manuscript by Nov 11 2024 11:59PM. If you will need more time than this to complete your revisions, please reply to this message or contact the journal office at plosone@plos.org. Please include the following items when submitting your revised manuscript:A rebuttal letter that responds to each point raised by the academic editor and reviewer(s). You should upload this letter as a separate file labeled 'Response to Reviewers'.A marked-up copy of your manuscript that highlights changes made to the original version. You should upload this as a separate file labeled 'Revised Manuscript with Track Changes'.An unmarked version of your revised paper without tracked changes. You should upload this as a separate file labeled 'Manuscript'.If applicable, we recommend that you deposit your laboratory protocols in protocols.io to enhance the reproducibility of your results. Protocols.io assigns your protocol its own identifier (DOI) so that it can be cited independently in the future. For instructions see: https://journals.plos.org/plosone/s/submission-guidelines#loc-laboratory-protocols. Additionally, PLOS ONE offers an option for publishing peer-reviewed Lab Protocol articles, which describe protocols hosted on protocols.io. Read more information on sharing protocols at https://plos.org/protocols?utm_medium=editorial-email&utm_source=authorletters&utm_campaign=protocols.

We look forward to receiving your revised manuscript.

Kind regards,

Nils Pilotte, Ph.D.

Academic Editor

PLOS ONE

Journal Requirements:

https://doi.org/10.1371/journal.pone.0288687

In your revision ensure you cite all your sources (including your own works), and quote or rephrase any duplicated text outside the methods section. Further consideration is dependent on these concerns being addressed.

3. Thank you for stating the following in the Competing Interest section: 

“RJN, DB, SD, and SC are Cepheid employees. Other authors have no competing interests.”

We note that one or more of the authors are employed by a commercial company: Cepheid 

2) Please also provide an updated Competing Interests Statement declaring this commercial affiliation along with any other relevant declarations relating to employment, consultancy, patents, products in development, or marketed products, etc.  

Within your Competing Interests Statement, please confirm that this commercial affiliation does not alter your adherence to all PLOS ONE policies on sharing data and materials by including the following statement: ""This does not alter our adherence to  PLOS ONE policies on sharing data and materials.” (as detailed online in our guide for authors http://journals.plos.org/plosone/s/competing-interests) . If this adherence statement is not accurate and  there are restrictions on sharing of data and/or materials, please state these. Please note that we cannot proceed with consideration of your article until this information has been declared.

4. We noted in your submission details that a portion of your manuscript may have been presented or published elsewhere:

“Yes, we utilize the a single sample from our letter to the editor (DOI 10.5588/ijtldopen.24.0124), however, that letter does not contain the details on the methodology we describe in this protocol paper.”

5. Please note that your Data Availability Statement is currently missing the repository name. If your manuscript is accepted for publication, you will be asked to provide these details on a very short timeline. We therefore suggest that you provide this information now, though we will not hold up the peer review process if you are unable.

7. Please amend either the title on the online submission form (via Edit Submission) or the title in the manuscript so that they are identical.

8. Upon checking your manuscript, we noticed that the reference list is not included on your paper, please provide the reference list at the end of your manuscript.

9. We note you have not yet provided a protocols.io PDF version of your protocol and/or a protocols.io DOI. When you submit your revision, please provide a PDF version of your protocol as generated by protocols.io (the file will have the protocols.io logo in the upper right corner of the first page) as a Supporting Information file. The filename should be S1_file.pdf, and you should enter “S1 File” into the Description field. Any additional protocols should be numbered S2, S3, and so on. Please also follow the instructions for Supporting Information captions [https://journals.plos.org/plosone/s/supporting-information#loc-captions]. The title in the caption should read: “Step-by-step protocol, also available on protocols.io.”

Please assign your protocol a protocols.io DOI, if you have not already done so, and include the following line in the Materials and Methods section of your manuscript: “The protocol described in this peer-reviewed article is published on protocols.io (https://dx.doi.org/10.17504/protocols.io.[...]) and is included for printing purposes as S1 File.” You should also supply the DOI in the Protocols.io DOI field of the submission form when you submit your revision.

If you have not yet uploaded your protocol to protocols.io, you are invited to use the platform’s protocol entry service [https://www.protocols.io/we-enter-protocols] for doing so, at no charge. Through this service, the team at protocols.io will enter your protocol for you and format it in a way that takes advantage of the platform’s features. When submitting your protocol to the protocol entry service please include the customer code PLOS2022 in the Note field and indicate that your protocol is associated with a PLOS ONE Lab Protocol Submission. You should also include the title and manuscript number of your PLOS ONE submission.

Reviewers' comments:

Reviewer's Responses to Questions

**Comments to the Author**

1. Does the manuscript report a protocol which is of utility to the research community and adds value to the published literature?

Reviewer #1: Yes

Reviewer #2: Yes

2. Has the protocol been described in sufficient detail?

To answer this question, please click the link to protocols.io in the Materials and Methods section of the manuscript (if a link has been provided) or consult the step-by-step protocol in the Supporting Information files.

The step-by-step protocol should contain sufficient detail for another researcher to be able to reproduce all experiments and analyses.

Reviewer #1: Partly

Reviewer #2: Partly

3. Does the protocol describe a validated method?

Reviewer #1: Yes

Reviewer #2: Yes

4. If the manuscript contains new data, have the authors made this data fully available?

Reviewer #1: Yes

Reviewer #2: Yes

**5. Is the article presented in an intelligible fashion and written in standard English?**

Reviewer #1: Yes

Reviewer #2: Yes

6. Review Comments to the Author

Reviewer #1: Overall this is a clever method for increasing the length of short read amplicon sequencing, while sustaining current sample input and sequencing reagent requirements. Although the authors focus on conferring mutations in pathogens relevant to human health, the described technique could be broadly applied to amplicon sequencing strategies requiring additional length beyond the traditional methods.

Recommendations for revision:

-The intro states the short read limitation of Illumina sequencing as 500 bp or 250x2 PE reads. Illumina offers 600 bp/ 300x2 PE read kits for MiSeq and NextSeq 1k/2k.

-It would be helpful to know the total bp size required for the current technique used to confer resistance mutations in TB.

-The provided GitHub link, www.github.com/SemiQuant/stilPCR, leads to a 404 error.

-"Controlling the concentrations of the gene-specific primers allows us to produce the

different amplicons at similar concentrations" How was this validated? Determined by the number of reads per primer, qPCR?

-Are the primer sequences used available? I was unable to find them in the manuscript, Protocols.IO page, or SI. Please include them or where to find them.

Reviewer #2: In this work, the authors describe a targeted amplicon sequencing protocol that generates both full-length and partial-length amplicons spanning a chosen genomic location. This is accomplished using hemi-nested PCR and results in the ability to sequence across genomic locations longer than would be possible using paired-end short read sequencing on an Illumina platform. The authors demonstrate the utility of the protocol on patient sputum by amplifying, sequencing, and mapping reads across a target on the Mycobacterium tuberculosis genome used for identifying resistance to a particular antibiotic. This method is very useful for analyses on samples for which there exists a very limited quantity since template for only one reaction tube is needed to amplify the longer genomic regions. In addition, although long read sequencing can be accomplished using other sequencing technologies, employing Illumina sequencing technology is important for variant calling because of the higher sequence quality obtained. The technique is well-explained, and the manuscript is well-written, however a few minor edits will bring valuable clarity for readers.

1. In the Expected Results section, only one example is depicted. While the figure clearly demonstrates the results, including only one example does not give information for readers to assess the reproducibility of the technique. Inclusion of another example figure along with a sentence regarding reproducibility would be helpful.

2. In the Protocol, the first table in Stage 1 PCR does not list concentrations. Concentrations of the primers and BSA should be included.

3. In the Protocol, the total volume for Stage 2 PCR is given as 55 uL. Adding the additional 4.5 uL listed would result in a total volume of 54.5 uL. Is the table missing a component or is the 55 uL obtained by rounding up?

7. PLOS authors have the option to publish the peer review history of their article (what does this mean?). If published, this will include your full peer review and any attached files.

Reviewer #1: No

Reviewer #2: No

---

## [Author Response · Author response to Decision Letter 0]

15 Oct 2024

PONE-D-24-37091

stilPCR increases the effective sequencing length of Illumina, targeted next-generation sequencing.

PLOS ONE

Thank you for the opportunity to submit a revised protocol and we thank the reviewers for their comments and suggestions, answering them has strengthened the manuscript. We have responded to the specific concerns below (in blue text) and made the relevant changes to the manuscript and protocol. Thank you for your consideration, and we look forward to hearing from you.

Reviewers' comments:

Reviewer's Responses to Questions

Comments to the Author

1. Does the manuscript report a protocol which is of utility to the research community and adds value to the published literature?

Reviewer #1: Yes

Reviewer #2: Yes

2. Has the protocol been described in sufficient detail?

To answer this question, please click the link to protocols.io in the Materials and Methods section of the manuscript (if a link has been provided) or consult the step-by-step protocol in the Supporting Information files.

The step-by-step protocol should contain sufficient detail for another researcher to be able to reproduce all experiments and analyses.

Reviewer #1: Partly

Reviewer #2: Partly

3. Does the protocol describe a validated method?

Reviewer #1: Yes

Reviewer #2: Yes

4. If the manuscript contains new data, have the authors made this data fully available?

Reviewer #1: Yes

Reviewer #2: Yes

5. Is the article presented in an intelligible fashion and written in standard English?

Reviewer #1: Yes

Reviewer #2: Yes

6. Review Comments to the Author

Reviewer #1: Overall this is a clever method for increasing the length of short read amplicon sequencing, while sustaining current sample input and sequencing reagent requirements. Although the authors focus on conferring mutations in pathogens relevant to human health, the described technique could be broadly applied to amplicon sequencing strategies requiring additional length beyond the traditional methods.

Recommendations for revision:

-The intro states the short read limitation of Illumina sequencing as 500 bp or 250x2 PE reads. Illumina offers 600 bp/ 300x2 PE read kits for MiSeq and NextSeq 1k/2k.

Thanks, we have updated this in the introduction. 

-It would be helpful to know the total bp size required for the current technique used to confer resistance mutations in TB.

We have clarified this in the expected results. “We have successfully used stilPCR to amplify two targets, atpE, and rv0678, of Mycobacterium tuberculosis in patient sputum[1]. rv0678 is a 498bp gene and mutations along its entirety may confer M. tuberculosis bedaquiline resistance[2,3].”

-The provided GitHub link, www.github.com/SemiQuant/stilPCR, leads to a 404 error.

-"Controlling the concentrations of the gene-specific primers allows us to produce the

different amplicons at similar concentrations" How was this validated? Determined by the number of reads per primer, qPCR?

Sorry, we had the GitHub repository set to private, we have now made it public. We have also added the following to claify our statement regarding the primer concentrations. 

“The tail sequences allow for even amplification of tailed products of different sizes. Controlling the concentrations of the gene-specific primers will allow you to produce the different amplicons at similar concentrations. The specific concentration of primer required can be determined by adding 0.5ul of 2000X SYBR green dye to the PCR reaction and monitoring the Ct values of the single plex reactions using a qPCR thermocycler. You can also control the maximum number of amplicons for Illumina cluster generation (sequencing) and ensure the maximum amount of generated data per sample by limiting our universal-tailed primers. Importantly, stilPCR-generated libraries can be pooled with any standard Illumina library, allowing for batching even with other sample types.”

-Are the primer sequences used available? I was unable to find them in the manuscript, Protocols.IO page, or SI. Please include them or where to find them.

Apologies, we have added them to the materials section of the protocol.

Reviewer #2: In this work, the authors describe a targeted amplicon sequencing protocol that generates both full-length and partial-length amplicons spanning a chosen genomic location. This is accomplished using hemi-nested PCR and results in the ability to sequence across genomic locations longer than would be possible using paired-end short read sequencing on an Illumina platform. The authors demonstrate the utility of the protocol on patient sputum by amplifying, sequencing, and mapping reads across a target on the Mycobacterium tuberculosis genome used for identifying resistance to a particular antibiotic. This method is very useful for analyses on samples for which there exists a very limited quantity since template for only one reaction tube is needed to amplify the longer genomic regions. In addition, although long read sequencing can be accomplished using other sequencing technologies, employing Illumina sequencing technology is important for variant calling because of the higher sequence quality obtained. The technique is well-explained, and the manuscript is well-written, however a few minor edits will bring valuable clarity for readers.

1. In the Expected Results section, only one example is depicted. While the figure clearly demonstrates the results, including only one example does not give information for readers to assess the reproducibility of the technique. Inclusion of another example figure along with a sentence regarding reproducibility would be helpful.

We have added the following to the expected results section: “Further examples demonstrating the reproducibility of stilPCR are available in Limberis et al[1].”

2. In the Protocol, the first table in Stage 1 PCR does not list concentrations. Concentrations of the primers and BSA should be included.

We have added all the concentrations of the reagents used.

3. In the Protocol, the total volume for Stage 2 PCR is given as 55 uL. Adding the additional 4.5 uL listed would result in a total volume of 54.5 uL. Is the table missing a component or is the 55 uL obtained by rounding up?

Apologies, we have corrected the volume of water in the first step, it was 16.5ul but should be 17ul.

7. PLOS authors have the option to publish the peer review history of their article (what does this mean?). If published, this will include your full peer review and any attached files.

Do you want your identity to be public for this peer review? For information about this choice, including consent withdrawal, please see our Privacy Policy.

Reviewer #1: No

Reviewer #2: No

---

## [Decision Letter · Decision Letter 1]

8 Nov 2024

stilPCR increases the effective sequencing length of Illumina, targeted next-generation sequencing.

PONE-D-24-37091R1

Dear Dr. Limberis,

We’re pleased to inform you that your manuscript has been judged scientifically suitable for publication and will be formally accepted for publication once it meets all outstanding technical requirements.

Kind regards,

Nils Pilotte, Ph.D.

Academic Editor

PLOS ONE

Additional Editor Comments (optional):

Reviewers' comments:

Reviewer's Responses to Questions

**Comments to the Author**

1. Does the manuscript report a protocol which is of utility to the research community and adds value to the published literature?

Reviewer #1: Yes

Reviewer #2: Yes

2. Has the protocol been described in sufficient detail?

To answer this question, please click the link to protocols.io in the Materials and Methods section of the manuscript (if a link has been provided) or consult the step-by-step protocol in the Supporting Information files.

The step-by-step protocol should contain sufficient detail for another researcher to be able to reproduce all experiments and analyses.

Reviewer #1: Yes

Reviewer #2: Yes

3. Does the protocol describe a validated method?

Reviewer #1: Yes

Reviewer #2: Yes

4. If the manuscript contains new data, have the authors made this data fully available?

Reviewer #1: Yes

Reviewer #2: Yes

**5. Is the article presented in an intelligible fashion and written in standard English?**

Reviewer #1: Yes

Reviewer #2: Yes

6. Review Comments to the Author

Reviewer #1: The authors made the suggested edits to the manuscript. My recommendation is to publish this work with no further edits.

Reviewer #2: The authors have increased the clarity of the manuscript by addressing all concerns and I have no further comments.

7. PLOS authors have the option to publish the peer review history of their article (what does this mean?). If published, this will include your full peer review and any attached files.

Reviewer #1: No

Reviewer #2: No

---

## [Editor Report · Acceptance letter]

28 Nov 2024

PONE-D-24-37091R1 

PLOS ONE

Dear Dr. Limberis, 

I'm pleased to inform you that your manuscript has been deemed suitable for publication in PLOS ONE. Congratulations! Your manuscript is now being handed over to our production team.

Kind regards, 

on behalf of

Dr. Nils Pilotte 

Academic Editor

PLOS ONE